# The Impact of Green Human Resource Management Practices on Employees, Clients, and Organizational Performance: A Literature Review

Aisha AlKetbi and John Rice *

College of Business Administration, University of Sharjah, Sharjah 500001, United Arab Emirates; u20102562@sharjah.ac.ae
* Correspondence: jrice@sharjah.ac.ae

**Abstract:** This literature review aims to examine the relationship between Green Human Resource Management (G-HRM) practices and various outcomes, including employee green attitudes, employee green satisfaction, client green satisfaction, employee green behavior, and organizational green performance. We reviewed existing literature on G-HRM practices and their impact on the selected outcomes. The review process involved the identification of articles through a systematic search in Scopus and Web of Science databases from January 2013 to December 2023. The search retrieved 2142 citations; of them, a total of 17 articles were deemed eligible for this review. The reviewed literature provides good evidence supporting a positive predictive relationship between G-HRM practices and employee green attitudes, employee green satisfaction, client green satisfaction, employee green behavior, and organizational green performance. However, there is a notable gap in studies exploring the influence of G-HRM practices on employee and client satisfaction. Overall, G-HRM practices emerge as a crucial tool for fostering environmentally conscious attitudes and behaviors among employees, ultimately contributing to enhanced employee satisfaction and improved organizational ecological performance. Future research should pay attention to the mechanisms underlying these relationships and explore potential moderating factors to enrich our understanding of the interrelated dynamics between G-HRM practices and sustainable outcomes.

**Keywords:** green human resource management (G-HRM); environmental sustainability; organizational performance

## 1. Introduction

The growing concern for environmental sustainability and corporate social responsibility has prompted organizations to adopt alternative avenues to meet these responsibilities (Martin et al. 2014). As emphasized by the United Nations General 17 Sustainable Development Goals (SDGs), organizations need to operationalize and integrate sustainability in their strategy, plans, and actions. Therefore, they can address the current and forthcoming stakeholder needs and ensure a better and sustainable future for all, balancing economic, social, and environmental development (Fonseca et al. 2020). Moreover, academic research (Leal Filho et al. 2023) highlights the need for stronger action to avoid major challenges such as climate catastrophe. One way to foster sustainable practices can be the application of Green Human Resource Management (G-HRM) (Pham et al. 2020). G-HRM, as a general term, refers to the integration of environmentally friendly practices into HRM strategies and policies (Ren and Hussain 2022). It is increasingly recognized as a means to enhance an organization's environmental performance and overall effectiveness (Munawar et al. 2022).

The adoption of G-HRM policies represents a proactive approach by agencies to enhance their environmental and overall performance. By integrating G-HRM practices into HRM strategies, organizations aim to foster positive attitudes, behaviors, and overall performance (Farrukh et al. 2022). Previous studies of the impact of G-HRM policies

on employee and organizational performance in terms of green attitudes, satisfaction, behaviors, environmental performance, and client satisfaction provided promising results. The body of literature examining the impact of G-HRM practices on organizational and employee performance underscores the pivotal role that environmentally conscious human resource practices play in shaping both individual and organizational outcomes (Faisal 2023). Scholars such as Ren et al. (2018) assert that G-HRM practices significantly influence employee attitudes and behaviors toward environmental sustainability, fostering a sense of ecological responsibility among employees (Ren et al. 2018). Additionally, research by Albloush et al. (2022) suggests a significant relationship exists between G-HRM and organizational performance, as well as between G-HRM and human capital development, demonstrating that integrating sustainability into human resource strategies can enhance an organization's environmental standing (Albloush et al. 2022). These findings align with insights from other works, such as those by (Baykal et al. 2023) and (Shafaei et al. 2020), which highlight the positive impact of G-HRM on employee job satisfaction and commitment to green initiatives. However, the literature also reveals nuanced outcomes, as observed in the research by (Aggarwal and Agarwala 2023), which identifies the mediating role of organizational culture in the effectiveness of G-HRM practices.

The aforementioned research provides a foundation for understanding the intricate relationships between G-HRM practices and organizational and employee performance, calling for further research and tailored implementation strategies to maximize their impact in diverse organizational contexts (Liao et al. 2011). A preliminary search of the literature determined that there has not been a review published on the topic of G-HRM practices in relation to employees, clients, and organizational behavior. It is this gap that we seek to address in this paper.

The rationale for this current study is grounded in the realization that the workforce plays a fundamental role in shaping societal values, norms, and practices (Amrutha and Geetha 2023; Brandis et al. 2017). As stewards of public resources, entities are accountable for the efficient delivery of services and are increasingly expected to lead by example in embracing sustainable practices (Faisal 2023). With its focus on integrating environmental considerations into human resource policies, G-HRM acts as a potential catalyst for fostering organizational cultures that are socially responsible and environmentally positive (Benevene and Buonomo 2020).

By critically examining existing research, this literature review aims to provide a comprehensive understanding of the extent to which G-HRM practices influence various dimensions of organizational and employee performance. The specific focus on employee green attitudes, green satisfaction, client green satisfaction, green behaviors, and organizational green performance reflects a multifaceted approach to evaluating the complex impact of G-HRM. Through this investigation, the research seeks to contribute evidence-based insights that can inform the design and implementation of effective G-HRM practices, fostering a more sustainable and responsible organizational culture.

Furthermore, the findings of this literature review are expected to have practical implications for organizations, policymakers, and researchers. The identification of key relationships and potential gaps in existing knowledge can guide organizational leaders in adopting and refining G-HRM strategies that resonate with the unique challenges and opportunities presented across various organizational contexts (Majid et al. 2023). Ultimately, this research endeavors to advance the discourse on sustainable human resource management, facilitating the integration of green practices that contribute to organizational success and broader environmental stewardship.

Therefore, the present literature review aims to critically examine existing research to determine the extent to which G-HRM practices positively predict employee green attitudes, employee green satisfaction, client green satisfaction, employee green behavior, and organizational green performance. Understanding these predictive relationships is crucial for organizations seeking to align their human resource practices with environmental

sustainability goals. Specifically, this literature review aims to address the following research question:

Do G-HRM policies positively predict:

(1)  Employee green attitudes and perceptions.
(2)  Employee green satisfaction.
(3)  Employee green behavior.
(4)  Client green satisfaction.
(5)  Organizational green performance.

## 2. Methods

### 2.1. Search Strategy

A comprehensive literature search was conducted across two main academic databases, namely Scopus and Web of Science. These databases were selected based on the recommendations in previous literature reviews (Faisal 2023). The Scopus and Web of Science databases have comprehensive coverage compared to other databases, and thus, the possibility of missing relevant articles is low (Norris and Oppenheim 2007). Therefore, only articles indexed in Scopus and Web of Science were considered in this review to ensure the quality of the findings. Furthermore, the reference lists of all the included articles were checked for possible relevant citations.

The search strategy employed a combination of keywords, Boolean operators, and subject headings related to "Green Human Resource Management", "sustainable human resource management", "organizational performance", "employee green behavior", "client satisfaction", and "environmental performance". By considering these concepts in our search strategy, we echo the recommendations of relevant literature, such as Tranfield et al. (2003), to underscore the importance of a systematic approach in conducting the literature review (Tranfield et al. 2003). The search was limited to articles published between January 2013 and December 2023 in English. The search strategy was designed to capture studies that directly addressed the relationship between G-HRM practices and employees, clients, or organizational behaviors.

### 2.2. Study Selection

The criteria for selecting papers were established based on the relevance of their content to our research questions outlined in the introduction. Specifically, we included studies that investigated the relationship between G-HRM practices and outcomes related to employees (e.g., job satisfaction), clients (e.g., satisfaction), and organizational performance. We excluded studies that did not directly address these relationships or those that lacked empirical evidence.

To ensure reliability and consistency in our review process, we employed a systematic approach guided by established guidelines for conducting literature reviews such as Tranfield et al. (2003). Two independent reviewers screened the identified articles based on predefined inclusion and exclusion criteria. Any discrepancies in article selection were resolved through discussion and consensus between the reviewers. Additionally, we utilized standardized data extraction forms to systematically extract relevant information from each included study, including data collection methods, study outcomes, and key findings.

The initial search yielded 2142 articles. Following the removal of 1749 duplicates, the titles and abstracts of 393 articles were screened for relevance to the research question. Articles that did not pertain to the G-HRM practices or policies were excluded during this phase. The full texts of 20 articles were then assessed for eligibility based on predetermined inclusion and exclusion criteria. Finally, 17 articles were determined eligible for this review and were further analyzed. Figure 1 presents the different steps of article screening and exclusion criteria.

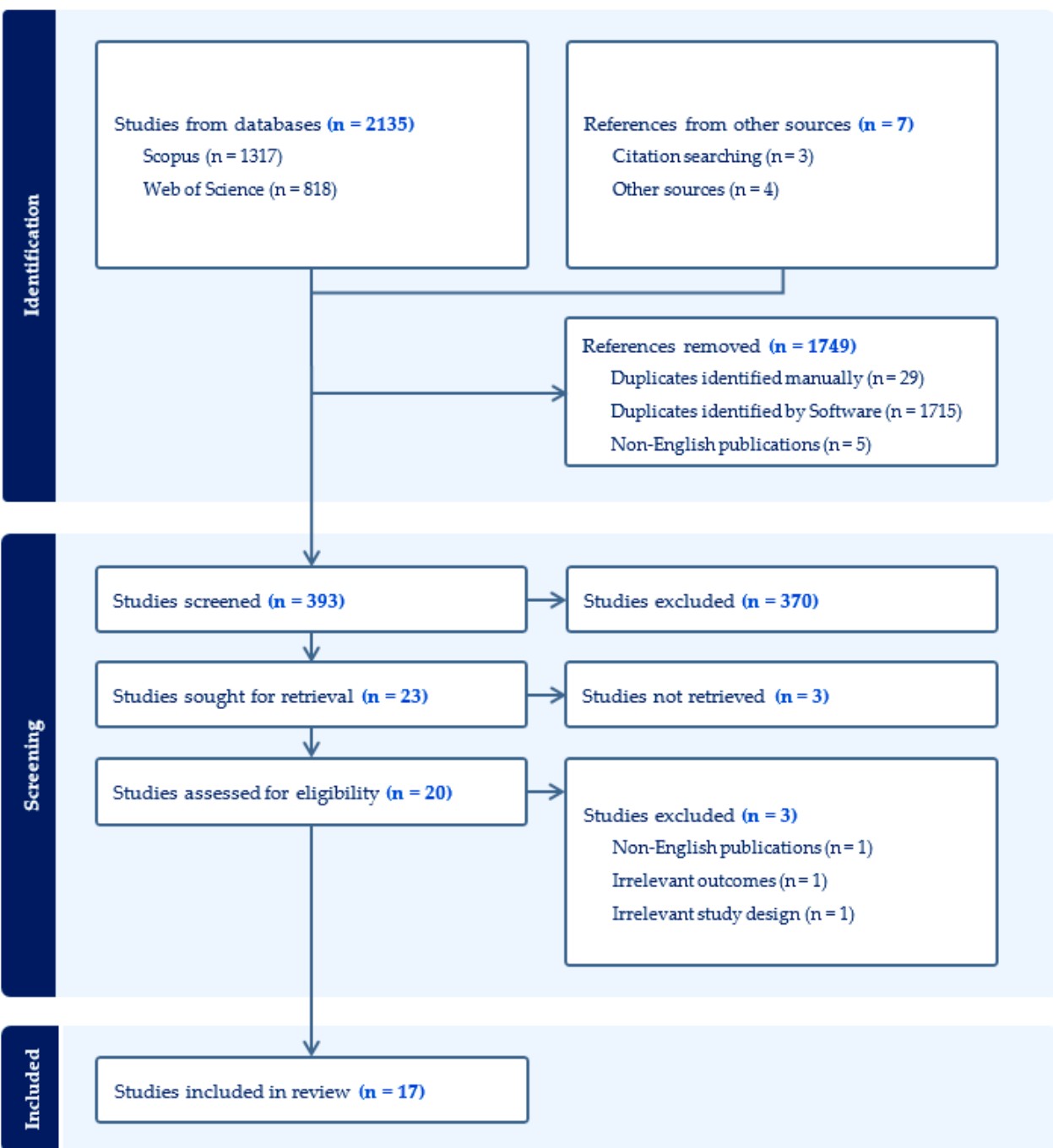

**Figure 1.** The screening framework.

*2.3. Inclusion Criteria*

This review considered studies that reported empirical findings on G-HRM practices and were conceptually based on valid theoretical frameworks. To be included, articles had to have been published in English in peer-reviewed journals that directly investigated the relationship between G-HRM practices and employees, clients, or organizational behaviors. To optimize the scope of this review, all study designs reporting primary empirical research, including quantitative or mixed-methods studies, were included. Similar literature reviews, including systematic reviews and data synthesis, were reviewed to provide background and context for the current study, but these articles were not included in the final analysis.

### 2.4. Exclusion Criteria

This review excluded studies that did not consider G-HRM as the main outcome. Articles lacking sufficient information to assess the impact of G-HRM practices on employees, clients, or organizational behavior were excluded. Also, this review excluded non-English publications or articles with an opinion or perspective nature, such as perspective reports, opinion papers, and editorials. Lastly, grey literature, including conference proceedings, organizations' websites, books, and book chapters, were excluded from this review.

### 2.5. Data Extraction and Synthesis

In the context of this study, data extraction and synthesis constitute a critical phase that involves systematically collecting and organizing information from selected studies to derive meaningful insights. This process typically involves the creation of a structured framework to capture essential data elements, such as author names, publication dates, the main research objectives, geographic location, participants, and empirical results. Subsequently, the synthesized data are systematically analyzed and interpreted to identify patterns, trends, and overarching themes within the literature. The synthesis phase involves a comprehensive examination of the gathered information, allowing the researcher to draw connections, identify gaps, and present a cohesive narrative that reflects the state of knowledge on the study topic. This systematic approach ensures the robustness and reliability of this literature review, providing readers with a well-informed and comprehensive understanding of the subject matter.

### 2.6. Theoretical Base of This Review

According to the above-mentioned criteria and the overall objectives, the researchers considered the Ability–Motivation–Opportunity (AMO) theory as a conceptual structure guiding this review (Appelbaum 2000). This conceptual framework seeks to understand and explain individual and organizational performance by considering three key factors: ability, motivation, and opportunity. Developed in the field of organizational behavior and management, the AMO theory posits that these three elements interact dynamically to influence an individual's or an organization's overall performance (Shoaib et al. 2021). Ability refers to the skills, knowledge, and competencies possessed by individuals; motivation encompasses the internal and external forces that drive behavior; and opportunity involves the external conditions or resources that facilitate or hinder performance (Appelbaum 2000). The AMO framework recognizes the interdependence of these factors, emphasizing that optimal performance occurs when there is a harmonious alignment of ability, motivation, and opportunity. This theory has practical applications in organizational development, performance management, and human resources research, offering a comprehensive perspective on enhancing productivity and achieving organizational goals (Wood and Horwitz 2015).

### 2.7. Definitions

In the context of this research, G-HRM practices refer to a set of environmentally conscious strategies that organizations adopt to align human resource functions with sustainability goals (Zaid et al. 2018). Specifically, G-HRM practice refers to the collection of regulations and policies that govern all green activities within institutional frameworks (Saeed et al. 2019). The employee life cycle within G-HRM is an organizational approach to visualize the degree of employee engagement within a given institution (Khan et al. 2022). This engagement includes integrating eco-friendly principles from recruitment to retirement, emphasizing the importance of environmental responsibility at every stage (Zhang et al. 2019). Rewards systems are designed to recognize and incentivize green behaviors, promoting a culture of sustainability within the workforce (Ahmad 2015). Education and training programs focus on enhancing employees' environmental awareness and skills, enabling them to contribute to the organization's sustainability objectives. Employee empowerment in G-HRM involves fostering a sense of responsibility and involvement in sustainable initiatives, encouraging employees to participate in green practices actively

(Tirno et al. 2023). Manager involvement is crucial in leading and supporting green HR practices, ensuring sustainability is integrated into organizational strategies and decision-making processes (Tuan 2022). Overall, G-HRM practices aim to create a holistic and environmentally responsible workplace by addressing various aspects of the employee experience.

In the realm of G-HRM, employee attitudes and perceptions refer to individuals' beliefs and views regarding environmental sustainability within the workplace (Garavan et al. 2023). It involves cultivating a positive environmental mindset among employees, emphasizing the importance of eco-conscious behavior (Chen et al. 2021). In the context of G-HRM, employee satisfaction is the extent to which employees are content with the organization's commitment to sustainability practices (Shafaei et al. 2020). This satisfaction is closely linked to employee behavior, as G-HRM encourages environmentally responsible actions and habits among staff, fostering a collective commitment to ecological stewardship (Abdelhamied et al. 2023). On the other hand, client satisfaction in G-HRM is the degree to which customers or clients perceive and appreciate an organization's sustainable initiatives, indicating a growing awareness and demand for environmentally friendly products and services (Wikhamn 2019). Lastly, within the scope of G-HRM, organizational performance is measured by the successful integration and implementation of green practices throughout the business, demonstrating a commitment to environmental responsibility that positively influences employee attitudes, client satisfaction, and overall success (Merlin and Chen 2022).

### 3. Results

A total of 17 studies were systematically reviewed to investigate the various dimensions of G-HRM practices. The selected studies encompassed diverse research designs, including mixed methods and cross-sectional analyses. Out of the 17 reviewed studies, two were mixed methods research (Masri and Jaaron 2017; Mousa and Othman 2020), while 15 used cross-sectional methods (Albloush et al. 2022; Ali and Hassan 2023; Al-Swidi et al. 2021; Arshad et al. 2022; Cahyadi et al. 2023; El Baroudi et al. 2023; Elshaer et al. 2023; Freire and Pieta 2022; Hameed et al. 2019; Li et al. 2023; Mensah et al. 2023; Rawashdeh 2018; Ren and Hussain 2022; Xiao et al. 2022; Zhang et al. 2019). The publication years of the included studies ranged from 2017 to 2023, highlighting a recent exploration of both contemporary and practical perspectives on G-HRM practices (See Figure 2). The geographical distribution of the studies was broad, with contributions from Europe, Asia, and the Middle East, providing a global perspective on the subject matter (See Figure 3).

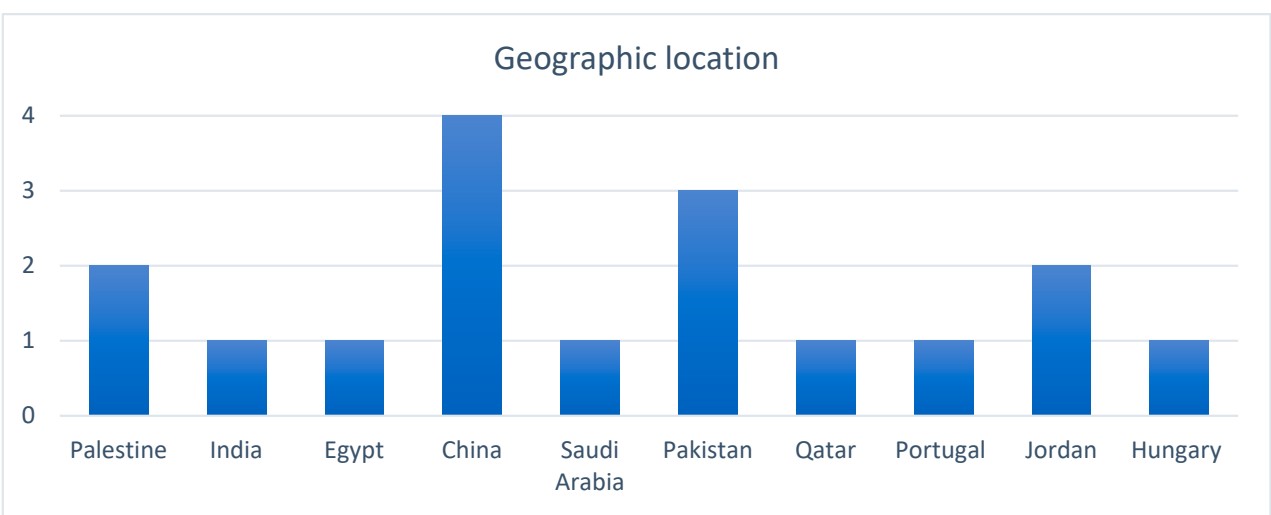

**Figure 2.** Distribution of studies by years of publication.

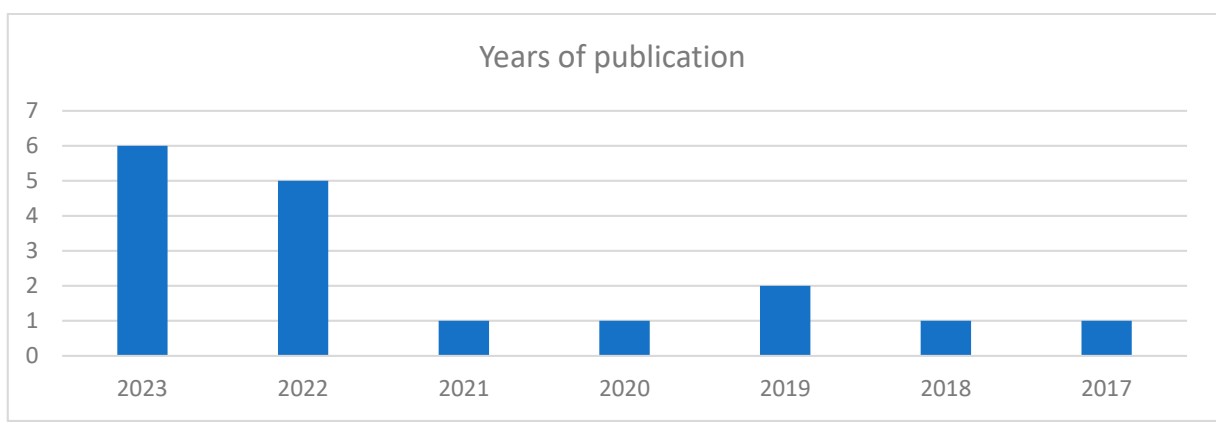

**Figure 3.** Distribution of studies by geographic location.

The sample sizes across the studies varied, with some involving large-scale cohorts exceeding 500 participants (Al-Swidi et al. 2021; Arshad et al. 2022), while others adopted a more focused approach with smaller, targeted populations of less than 100 participants (Ali and Hassan 2023; Mousa and Othman 2020; Rawashdeh 2018). Methodologically, the studies employed a variety of tools to assess variables of interest, with online surveys (Albloush et al. 2022; Cahyadi et al. 2023; Freire and Pieta 2022; Rawashdeh 2018; Ren and Hussain 2022), face-to-face interviews (Masri and Jaaron 2017; Mousa and Othman 2020), and standardized paper-based questionnaires being prevalent (Ali and Hassan 2023; Al-Swidi et al. 2021; Arshad et al. 2022; El Baroudi et al. 2023; Elshaer et al. 2023; Hameed et al. 2019; Li et al. 2023; Mensah et al. 2023; Xiao et al. 2022; Zhang et al. 2019). The diverse characteristics of the included studies, as presented in Table 1, contribute to the richness and depth of the synthesized findings presented in this literature review.

**Table 1.** Study characteristics.

| Authors (Year) | Country | Aim of Study | Study Design | Population Description | Data Collection Method | Number of Participants |
|---|---|---|---|---|---|---|
| Ali and Hassan (2023) | Egypt | To assess nurse managers' perception about G-HRM practices. | Cross-sectional | Nurse managers in five University Hospitals | Paper-based questionnaire | 95 |
| Cahyadi et al. (2023) | Hungary | To examine relationship between G-HRM policies and employees' behavior | Cross-sectional | Prospective employees in business sector | E-Survey | 252 |
| El Baroudi et al. (2023) | China | To examine the role of G-HRM on employee green behavior and overall organizational performance | Cross-sectional | Team members working in 4 hospitality and tourism settings | Paper-based questionnaire | 277 |
| Elshaer et al. (2023) | Saudi Arabia | To examine the relationship between G-HRM and organizational performance and how employee pro-environmental behavior may moderate this relationship. | Cross-sectional | Employees at managerial level in small- and medium-sized hotels and travel agencies | Paper-based questionnaire | 304 |

| Authors (Year) | Country | Aim of Study | Study Design | Population Description | Data Collection Method | Number of Participants |
|---|---|---|---|---|---|---|
| Li et al. (2023) | China | To examine the effect of G-HRM practices on an employees' green behavior. | Cross-sectional | Employees working in Chinese Multinational Corporation | Paper-based questionnaire | 374 |
| Mensah et al. (2023) | India | To investigate the relationship between G-HRM practices and employee green behavior in the hospital settings. | Cross-sectional | Human resource and administrative managers at Korle Bu Teaching Hospital | Paper-based questionnaire | 264 |
| Albloush et al. (2022) | Jordan | To determine the effect of G-HRM policies on organizational performance | Cross-sectional | Employees in public institutions | E-survey | 275 |
| Arshad et al. (2022) | Pakistan | To examine the impact of G-HRM policies on employees' attitudes, satisfaction, and behavior. | Cross-sectional | Employees in the hospitality sector | Paper-based questionnaire | 508 |
| Freire and Pieta (2022) | Portugal | To analyze the impact of G-HRM on organizational behavior through the mediating role of organizational identification and job satisfaction | Cross-sectional | Employees in industrial companies | E-Survey | 120 |
| Ren and Hussain (2022) | Pakistan | Explores the direct and indirect effects of G-HRM on the environmental performance. | Cross-sectional | Employees in manufacturing companies. | E-survey | 306 |
| Xiao et al. (2022) | China | The study focuses on investigating the moderating role of G-HRM on consumer behavior and Employee Performance. | Cross-sectional | Frontline employees of the hospitality sector. | Paper-based questionnaire | 210 |
| Al-Swidi et al. (2021) | Qatar | To examine the effects of G-HRM culture on employees' behavior | Cross-sectional | Employees in public and private sector | Paper-based questionnaire | 632 |
| Mousa and Othman (2020) | Palestine | To assess the level of implementation of G-HRM practices in healthcare sector and their impact on sustainable performance. | Mixed Methods | HR experts from the healthcare sector | Interview | 69 |

**Table 1.** *Cont.*

| Authors (Year) | Country | Aim of Study | Study Design | Population Description | Data Collection Method | Number of Participants |
|---|---|---|---|---|---|---|
| Hameed et al. (2019) | Pakistan | To test the role of G-HRM practices on employee green behavior and to investigates the moderating effect of individual green values. | Cross-sectional | Employees and their supervisors in a large manufacturing company | Paper-based questionnaire | 365 |
| Zhang et al. (2019) | China | To examine the influence of five types of G-HRM practices (employee life cycle, rewards, education and training, employee empowerment, and manager involvement) on employee green behavior in the workplace. | Cross-sectional | Employees in different industrial settings. | Paper-based questionnaire | 145 |
| Rawashdeh (2018) | Jordan | To explore the relationship between G-HRM practices and environmental performance in Jordanian health service organization | Cross-sectional | Hospital managers | E-Survey | 91 |
| Masri and Jaaron (2017) | Palestine | To examine the effect of G-HRM practices on Environmental Performance | Mixed Methods | Organizations operating in manufacturing sectors | Interview | 110 |

The results of this literature review revealed a diverse range of study settings encompassed within the reviewed articles. The exploration of these scholarly works unveiled a broad spectrum of organizational settings, with a notable emphasis on the industrial sector (Freire and Pieta 2022; Hameed et al. 2019; Li et al. 2023; Masri and Jaaron 2017; Ren and Hussain 2022), the hospitality and tourism sector (Arshad et al. 2022; El Baroudi et al. 2023; Elshaer et al. 2023; Xiao et al. 2022), healthcare sector (Ali and Hassan 2023; Mensah et al. 2023; Mousa and Othman 2020; Rawashdeh 2018), enterprises (Cahyadi et al. 2023; Zhang et al. 2019), and public services (Albloush et al. 2022; Al-Swidi et al. 2021) (See Figure 4). The inclusion of such varied contexts not only underscores the multidimensionality of the research landscape but also emphasizes the predominant nature of the phenomena under investigation.

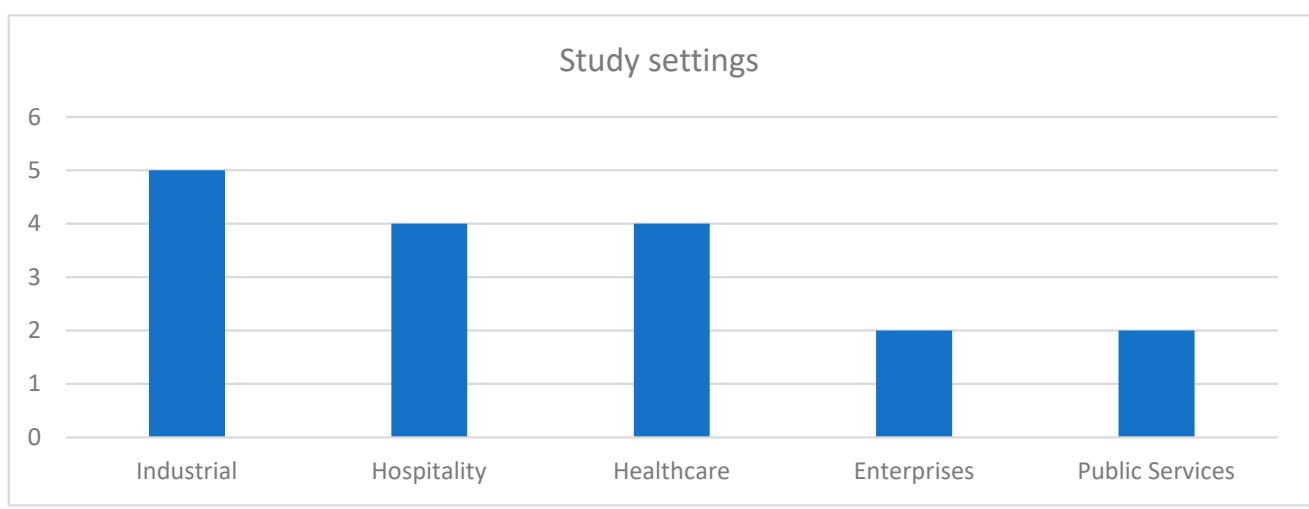

**Figure 4.** Distribution of studies by Setting.

Table 2 presents the findings concerning the study setting, relevant outcomes and summarizes the main results of the included studies. Overall, the main findings of the 17 reviewed articles suggest a positive significant correlation between G-HRM practices and employees, clients, or organizational performance. The following section presents these findings in detail.

**Table 2.** Presents study settings, outcomes studied, and the main findings.

| Authors (Year) | Setting | Study Outcomes | | | | | Main Findings |
| --- | --- | --- | --- | --- | --- | --- | --- |
| | | Employee Attitude | Employee Satisfaction | Employee Behavior | Client Satisfaction | Organizational Performance | |
| Ali and Hassan (2023) | Healthcare | √ | - | - | - | - | A statistically significant positive correlation exists between G-HRM Practices and employees' work values. |
| Cahyadi et al. (2023) | Enterprises | - | - | √ | - | - | G-HRM practices positively influence employees' green behavior. G-HRM practices mediate the relationship between green transformational leadership and employees' green behavior. |
| El Baroudi et al. (2023) | Hospitality and Tourism | √ | - | √ | - | √ | Green employee behaviors mediate the relationship between employee attitudes and perceptions of G-HRM and nongreen performance. |
| Elshaer et al. (2023) | Hospitality and Tourism | - | - | √ | - | √ | G-HRM practices can improve environmental, economic, and social performance, and these relationships can be strengthened through the moderating effects of employees' pro-environmental behavior. |

**Table 2.** *Cont.*

| Authors (Year) | Setting | Study Outcomes | | | | | Main Findings |
|---|---|---|---|---|---|---|---|
| | | Employee Attitude | Employee Satisfaction | Employee Behavior | Client Satisfaction | Organizational Performance | |
| Li et al. (2023) | Industry | - | - | √ | - | - | G-HRM practices have a positive effect on employees' green behavior. Psychological green climate mediates the relation between G-HRM practices and employee in-role green behavior. |
| Mensah et al. (2023) | Healthcare | - | - | √ | - | - | Green training, green hiring, and green compensation were significant predictors of innovative work behavior among employees. |
| Albloush et al. (2022) | Public Services | - | - | - | - | √ | A significant association between G-HRM (green rewards, compensation, and training) and organizational performance. Human Capital mediates the link between G-HRM and Organizational Performance. |
| Arshad et al. (2022) | Hospitality and Tourism | √ | √ | √ | - | - | Employee environmental attitudes encourage employees' ecological behavior and satisfaction with the organization. |
| Freire and Pieta (2022) | Industry | - | √ | - | - | √ | There is a mediation effect of job satisfaction on the relationship between G-HRM and its impact on organizational citizenship behavior. |
| Ren and Hussain (2022) | Industry | - | - | √ | - | √ | A positive and significant effect of G-HRM on employee and organizational environmental performance. There is a partial mediation of employee environmental performance. |
| Xiao et al. (2022) | Hospitality and Tourism | - | - | √ | √ | - | G-HRM, aka G-HRM, directly impacts consumer behavior. Diffidence moderates the relationship between G-HRM and employee performance and employee eco-friendly behavior. |

**Table 2.** *Cont.*

| Authors (Year) | Setting | Study Outcomes | | | | | Main Findings |
|---|---|---|---|---|---|---|---|
| | | Employee Attitude | Employee Satisfaction | Employee Behavior | Client Satisfaction | Organizational Performance | |
| Al-Swidi et al. (2021) | Public Services | - | - | √ | - | √ | A significant effect of G-HRM practices on green organizational culture. G-HRM practices has a significant positive relationship with employees' behavior and organizational performance. |
| Mousa and Othman (2020) | Healthcare | - | - | - | - | √ | G-HRM practices had a positive influence on sustainable performance |
| Hameed et al. (2019) | Industry | √ | - | √ | - | - | G-HRM has a significant indirect effect on employee organizational citizenship behavior toward environment through green employee empowerment. The individual green values moderated this relationship. |
| Zhang et al. (2019) | Enterprises | - | - | √ | - | - | G-HRM practices (employee life cycle, education and training, employee empowerment, and manager involvement) all had significant positive effect on the employees in-role and extra-role green behavior. |
| Rawashdeh (2018) | Healthcare | - | - | - | - | √ | A positive significant correlation between G-HMR practices (recruitment and selection, training, and development) and organizational performance. |
| Masri and Jaaron (2017) | Industry | - | - | - | - | √ | A positive significant correlation between the G-HRM practices (recruitment and selection, training and development, performance management and appraisal, reward and compensation, employee empowerment and participation, and green management) and environmental performance. |

As shown in Table 2, the comprehensive review of the literature revealed several recurring themes and trends within the research on G-HRM practices. A predominant finding across studies was the consistent positive correlation between G-HRM practices and employee behavior, as evidenced by Cahyadi et al. (2023), El Baroudi et al. (2023), Elshaer

et al. (2023), Li et al. (2023), Mensah et al. (2023), Arshad et al. (2022), Ren and Hussain (2022), Xiao et al. (2022), Al-Swidi et al. (2021), Hameed et al. (2019), and Zhang et al. (2019). The above-mentioned studies examined various G-HRM practices such as green training, green hiring, green compensation, and employee empowerment and reported a significant positive correlation with employee green behavior. These findings collectively suggest a significant predictive validity of G-HRM practices on employee green behavior, indicating that organizations implementing G-HRM practices are expected to achieve a higher level of employee green behavior. This highlights the value of adopting environmentally sustainable practices to foster a culture of green practices among employees in a given context.

Organizational performance is another theme that has been widely examined in previous studies as a significant outcome of implementing G-HRM practices (Albloush et al. 2022; Al-Swidi et al. 2021; El Baroudi et al. 2023; Elshaer et al. 2023; Freire and Pieta 2022; Masri and Jaaron 2017; Mousa and Othman 2020; Rawashdeh 2018; Ren and Hussain 2022). Elshaer et al. (2023) reported that the adoption of G-HRM practices has the potential to enhance several aspects of organizational performance, including environmental, economic, and social performance. Moreover, the positive connections established can be further reinforced by the moderating influence of employees' pro-environmental behavior (Elshaer et al. 2023). Similarly, Albloush et al. (2022) found a significant association between G-HRM practices such as green rewards, compensation, training, and organizational overall performance. They also found that human capital mediates the association between G-HRM practices and organizational performance (Albloush et al. 2022). Rawashdeh (2018) investigated slightly different aspects of G-HRM, including green recruitment and selection, green training, and development, and reported a significant positive correlation between these G-HMR practices and organizational performance. It is clear from the literature that G-HRM practices not only improved the organization's triple bottom line performance (i.e., environmental, economic, and social) but also can sustain a competitive advantage such as improvement of employees' and the local community's conditions (Elshaer et al. 2023).

Regarding employees' green attitudes, four studies (Ali and Hassan 2023; Arshad et al. 2022; El Baroudi et al. 2023; Hameed et al. 2019) affirm a positive correlation with G-HRM practices. Ali et al. (Ali and Hassan 2023) reported a statistically significant positive correlation between G-HRM Practices and employees' work values, defined as the beliefs and attitudes employees hold toward their work and the workplace. Ali et al. further argued that work values can influence employee motivation, job satisfaction, and commitment to the organization (Ali and Hassan 2023). El Baroudi et al. (2023) Took a different perspective to examine G-HRM practices and their influence on employees' behavior and attitudes and found that in-role and extra-role green employee behaviors mediate the relationship between employee attitudes and perceptions of G-HRM practices. Interestingly, Arshad et al. (2022) took a step further and explored the relationship between employees' attitudes on their job satisfaction and behavior and found that employee environmental attitudes encourage employees' ecological behavior and satisfaction with the organization. Hameed et al. (2019) studied the influence of G-HRM practices on employees' environmental performance and found that G-HRM practices significantly indirectly affect organizational citizenship behavior toward the environment through green employee empowerment. They also found that individual green values moderated this relationship (Hameed et al. 2019). Although these four studies mentioned above examined different G-HRM practices, such as eco-friendly training and the promotion of environmental values, there seems to be an agreement that G-HRM practices contribute to the cultivation of a green organizational culture, fostering positive attitudes toward sustainable practices among employees and in turn positive green and general performance outcomes.

Conversely, studies investigating the impact of G-HRM practices on employees' and clients' satisfaction found a notable scarcity in findings. Only one study investigated client satisfaction; two articles discussed employee satisfaction as the main outcome. While Arshad et al. (2022) and Freire and Pieta (2022) reported significant positive effects of G-

HRM practices on employees' satisfaction, Xiao et al. (2022) indicated that G-HRM directly impacts client satisfaction and behavior. Although research in this area is limited, emerging evidence from these studies suggests a positive association between G-HRM practices and employee and client green satisfaction. Organizations that prioritize environmental responsibility in HR practices are perceived positively by employees and clients, leading to increased customer satisfaction and strengthened business relationships.

## 4. Discussion

### 4.1. The Main Findings

G-HRM has gained significant attention as organizations seek sustainable practices. This literature review synthesizes findings from 17 key articles published between 2013 and 2023, exploring the impact of G-HRM practices on both organizational and employee performance. The results of this comprehensive literature review affirm the positive predictive relationship between G-HRM policies and employee green attitudes, employee green satisfaction, client green satisfaction, employee green behavior, and organizational green performance. This current study's findings agree with previous literature reviews that examined G-HRM practices (Benevene and Buonomo 2020; Faisal 2023). The overall synthesis of findings underscores the importance of G-HRM as a strategic tool for organizations aiming to enhance their environmental sustainability initiatives. This concordance highlights the significance of G-HRM practices as influential tools for fostering environmentally conscious attitudes and behaviors within organizational settings (Ren et al. 2018). The positive linkages identified across multiple dimensions emphasize the potential of G-HRM practices to enhance ecological performance and contribute to employee and client satisfaction (Xiao et al. 2022). However, a noteworthy observation arising from the literature review is the scarcity of studies specifically examining the influence of G-HRM practices on employee and client satisfaction. This gap in the literature suggests an area where future research could make meaningful contributions. Understanding how G-HRM practices directly impact the satisfaction levels of employees and clients is essential for comprehensively evaluating the holistic impact of these practices on organizational effectiveness and employees' work values.

Several studies (Ali and Hassan 2023; Al-Swidi et al. 2021; Arshad et al. 2022; Ren et al. 2018) highlighted the concept of 'work values' as an influential factor in implementing G-HRM practices. Wang et al. (2019) defined four subdomains of work values, including social (ideals that emphasize societal contributions), extrinsic (values linked to employment stability and compensation), prestige (values connected to power and influence), and intrinsic (ideals related to autonomy and development). The classification of work values into four distinct subdomains offers a nuanced understanding of individuals' priorities and beliefs in the workplace (Baykal et al. 2023). The social subdomain underscores the importance of ideals prioritizing societal contributions, reflecting a commitment to positively impacting the community through one's professional endeavors (Ali and Hassan 2023). In contrast, the extrinsic subdomain centers on values associated with job stability and compensation, encapsulating tangible and external factors that individuals deem significant in their work lives. The prestige subdomain goes into values tied to power and influence within the workplace, shedding light on the importance of status and recognition. Lastly, the intrinsic subdomain encapsulates ideals related to personal autonomy and professional development, emphasizing the intrinsic rewards derived from the nature and growth of one's work (Wang et al. 2019). This comprehensive framework provides a holistic perspective on the multifaceted nature of work values, encompassing societal contributions, external rewards, status, and personal fulfillment. It has been documented that G-HRM practices can optimize employees' work values, and this improvement in employees' work values potentially improves overall organizational performance (Liu and Lin 2020).

As organizations increasingly prioritize sustainability initiatives, G-HRM practices emerge as pivotal instruments in aligning environmental stewardship with employee satisfaction and ecological performance. The literature supports the idea that G-HRM

practices catalyze a workplace culture that values and integrates environmentally responsible behaviors (Arshad et al. 2022; El Baroudi et al. 2023; Li et al. 2023). Ren et al. (2018) noted that "designing and implementing G-HRM practices requires major investments in organizational resources, likely leading managers to question whether such investments are worthwhile." It is also imperative to underscore that implementing G-HRM practices is imperative rather than a luxury; thus, their planning, development, implementation, and follow-up are important responsibilities (Amrutha and Geetha 2023).

Future research endeavors should examine the mechanisms that drive the observed relationships. Investigating the specific pathways through which G-HRM practices influence employee attitudes, behaviors, and satisfaction, as well as their impact on client satisfaction, can provide actionable insights for organizations seeking to enhance their sustainability initiatives. Exploring potential moderating factors can also contribute to a nuanced understanding of the dynamics involved, allowing for more context-specific and effective implementation of G-HRM practices. Furthermore, considering the influence of the country's culture and the performance of the Sustainable Development Goals (SDGs), we can see that this has the potential to improve our understanding of the actual impact of all these factors on GHRM practices. Lastly, there may be a need to examine the different practices of G-HRM in specific contexts. For example, there seem to be limited studies that examined the G-HRM practices in healthcare settings (Ali and Hassan 2023; Mensah et al. 2023; Mousa and Othman 2020; Rawashdeh 2018). Examining G-HRM practices in the healthcare sector may add valuable insights and contribute evidence-based strategies that can inform the design and implementation of effective G-HRM practices, fostering a more sustainable and responsible hospital culture.

While the existing literature establishes a compelling case for the positive outcomes associated with G-HRM practices, there remains untapped potential for research to further elucidate the interplay between these practices and sustainable organizational outcomes. The recommendations stemming from this discussion underscore the need for future studies to address the identified gaps and advance our understanding of G-HRM practices in the context of fostering sustainability.

### 4.2. Academic, Managerial and Policy Implications

Academically, our review confirms the importance of G-HRM as a strategic tool for organizations aiming to enhance their environmental sustainability initiatives and performance more broadly. The review also finds the literature nascent, highlighting the importance of further research that may unpack the G-HRM "black box" to explore the mechanisms underlying the relationships between G-HRM practices, organizational sustainability performance, and employee and client satisfaction. As part of this academic project, the investigation of potential moderating and mediating relationships and better cultural and sectoral contextualization may provide valuable insights for developing context-specific and effective G-HRM strategies.

From a managerial perspective, we find that there is evidence that G-HRM practices can facilitate improved organizational performance across various dimensions, including environmental, economic, team/HR, and social elements. Managers can leverage G-HRM practices to enhance employee attitudes, behaviors, and satisfaction. This, in turn, may inculcate environmental responsibility within the wider organization.

In terms of policy implications, the positive relationships between G-HRM practices and organizational outcomes suggest the need for policy levers that promote the adoption of environmentally responsible HR practices by organizations. The encouragement of sustainability initiatives in HR policies can contribute to the national achievement of, for example, Sustainable Development Goals (SDGs) and Conférence des Parties (COP) goals. In addition, policymakers can support research programs to improve and understand the implementation of G-HRM practices across different sectors and cultural contexts, facilitating evidence-based policy development.

*4.3. Strengths and Limitations*

The literature review conducted in this study presents several notable strengths and acknowledges some limitations. A key strength lies in the comprehensive utilization of the Scopus and Web of Sciences databases, ensuring a broad coverage of relevant articles. The rigorous application of eligibility criteria further enhances the credibility of the review, focusing on high-quality literature. Additionally, the commitment to utilizing systematic article selection to the best of our knowledge contributes to a thorough analysis of the existing literature. However, the study is not without limitations. The exclusive reliance on English language publications may overlook valuable insights from other languages and cultural representations, potentially limiting the diversity of perspectives. The exclusion of non-journal sources, such as conference proceedings and books, could impact the depth of coverage. Moreover, despite efforts to be exhaustive, there is a possibility of overlooking some papers, highlighting the inherent challenge of achieving complete coverage in a dynamic research landscape. Lastly, the literature review did not encompass studies from Africa, America, and Oceania, which may require further exploration. Recognizing these strengths and limitations is crucial for carefully interpreting the findings.

**5. Conclusions**

In conclusion, the reviewed literature provides good evidence supporting the positive predictive relationship between G-HRM practices and employee green attitudes, employee green satisfaction, client green satisfaction, employee green behavior, and organizational green performance. However, limited studies have examined the influence of G-HRM practices on employee and client satisfaction, and there is a need to include these outcomes in future studies. As organizations prioritize sustainability, G-HRM practices emerge as a critical tool for fostering environmentally conscious attitudes and behaviors, thereby contributing to both employee satisfaction and organizational ecological performance. Future research should focus more on the mechanisms underlying these relationships and explore potential moderating factors to further enhance our understanding of the dynamics between G-HRM practices and sustainable outcomes.

**Author Contributions:** Conceptualization, A.A. and J.R.; methodology, A.A.; validation, A.A. amd J.R. formal analysis, A.A.; investigation, A.A.; resources, A.A.; data curation, A.A.; writing—original draft preparation, A.A.; writing—review and editing, J.R.; visualization, A.A.; supervision, J.R.; project administration, A.A. All authors have read and agreed to the published version of the manuscript.

**Funding:** This research received no external funding.

**Institutional Review Board Statement:** Not applicable.

**Informed Consent Statement:** Not applicable.

**Data Availability Statement:** No new data were created in this study. Data sharing is not applicable to this article.

**Conflicts of Interest:** The authors declare no conflicts of interest.

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
