# Peer review of "The Impact of Green Human Resource Management Practices on Employees, Clients, and Organizational Performance: A Literature Review"

_admsci, doi:10.3390/admsci14040078_

Round 1

Reviewer 1 Report

Comments and Suggestions for Authors

There is only one recommendation: in my opinion, it is recommended to better present (some additional information and examples, if possible) for the criteria used to assess the relevance of the selected literature (which would be some elements for assessing the relevance of an existing paper to the G-HRM topic) 

Author Response

Response to Reviewers’ Comments

Dear Editor and Reviewers,

We highly appreciate your time and efforts to review our manuscript. We carefully addressed all the concerns raised by the reviewers and addressed them in the revised manuscript. Our specific responses to the reviewers’ comments can be found below.

Sincerely,

Reviewer 1:

Comment C1:

In the introduction section, the author(s) should emphasize the relevance of

Sustainability and the urgency to adopt sustainable models and practices.

Response:

Thank you for your valuable suggestion. We revised the introduction based on your note. We added more information discussing the relevance of sustainability to the matter.

Comment C2:

The author(s) should reinforce the methodological dimension of the literature review. (e.g., "A literature review aims to reflect the state of knowledge in a specific subject supported by a methodical behavior (Tranfield et al., 2003)) and the choices of the selected keywords, considering the five proposed research questions. Moreover, please carefully explain the choice of the selected keywords. The keywords do not include terms such as "employee green behavior" or "client satisfaction", which might be a reason for the "scarcity of studies specifically examining the influence of G-HRM practices on employee and client satisfaction ". For the reviewer, this issue must be explained by the author(s).

Response:

We appreciate your insightful comment regarding the methodological dimension of the literature review and the selection of keywords.

We agree that reinforcing the methodological aspect of the literature review is crucial to ensure a robust representation of the state of knowledge in the field. We certainly emphasized this aspect more explicitly in the revised version of the manuscript, drawing upon relevant literature such as Tranfield et al. (2003) to underscore the importance of a systematic approach in conducting the literature review.

Regarding the choice of selected keywords, we acknowledge your concern about the absence of terms such as "employee green behavior" and "client satisfaction" in the methods of the submitted version of the manuscript because we only cited an example of the terms used in our database search. In fact, we used a long list of terms and keywords in our search, and we used different strategies to comprehensively explore the published literature using not only keywords, but also Boolean operators, and subject headings as we cited in the search strategy section.

To address your concern, we added all terms we used in our search including "employee green behavior" and "client satisfaction." These are indeed important concepts, and we understand how their inclusion could contribute to a more clarity and quality of our review.

Comment C3:

The author(s) claim in the introduction that this study has a special focus on the public sector, but this is not evident in the title and the remainder of the article.

Response:

Thank you for your valuable note. This was a typo error in the introduction as our review was not limited to only public sector. We revised the words in the introduction to align with the methods and results of this review.

Comment C4:

Please clarify the method and criteria for studying the selected papers and how reliability and consistency were ensured.

Response:

Thank you for your thoughtful comment regarding the methods and criteria employed in our literature review.

Methodologically, our approach involved a systematic search of two rigorous academic databases, namely, Scopus, and Web of Science, using predefined search terms related to G-HRM practices and their effects on employees, clients, and organizational performance. We limited our search to these two databases to ensure the quality of the reviewed studies. We only included peer-reviewed articles published within a specified timeframe to ensure the inclusion of recent and relevant literature. Additionally, we employed snowballing techniques, where references of included articles were also examined to identify additional relevant studies.

The criteria for selecting papers were established based on the relevance of their content to our research questions outlined in the introduction. Specifically, we included studies that investigated the relationship between G-HRM practices and outcomes related to employees (e.g., job satisfaction), clients (e.g., satisfaction), and organizational performance. We excluded studies that did not directly address these relationships or those that lacked empirical evidence.

To ensure reliability and consistency in our review process, we employed a systematic approach guided by established guidelines for conducting literature reviews such as Tranfield et al. (2003). Two independent reviewers screened the identified articles based on predefined inclusion and exclusion criteria. Any discrepancies in article selection were resolved through discussion and consensus between the reviewers. Additionally, we utilized standardized data extraction forms to systematically extract relevant information from each included study, including data collection methods, study outcomes, and key findings.

Lastly, we are confident that our literature review adheres to rigorous methodological standards, including systematic search strategies, predefined inclusion and exclusion criteria, and independent screening by multiple reviewers. We provided this explanation of our methodological approach and reliability assurance measures in the revised manuscript to address your concerns. (See section: 2.2 Study Selection).

Comment C5:

As a suggestion for future investigations, the author (s) could consider the influence of the country's culture and SDGs performance on the research results.

Response:

We appreciate your constructive feedback. We added your suggestion to the future research section.

Comment C6:

As a research limitation, the author (s) could add that the literature review did not encompass studies from Africa, America, and Oceania.

Response:

We appreciate your valuable note. To address your concern, we revised the limitations section and added your suggestions.

Comment C7:

Please develop the academic, managerial and policy implications for this study.

Response:

The following section was added:

4.2. Academic, managerial and policy implications

Academically, our review confirms the importance of G-HRM as a strategic tool for organizations aiming to both enhance their environmental sustainability initiatives, and their performance more broadly. The review also finds the literature in a nascent state, and it highlights the importance of further research that may unpack the G-HRM “black box” to explore the mechanisms underlying the relationships between G-HRM practices, or-ganizational sustainability performance, and employee and client satisfaction. As part of this academic project the investigation of potential moderating and mediating relation-ships, and better cultural and sectoral contextualization, may provide valuable insights for developing context-specific and effective G-HRM strategies.

From a managerial perspective, we find that there is evidence that G-HRM practices can facilitate improved organizational performance across various dimensions, including environmental, economic, team/HR and social elements. Managers can leverage G-HRM practices to enhance employee attitudes, behaviors, and satisfaction. This in turn may in-culcate environmental responsibility within the wider organization.

In terms of policy implications, the positive relationships between G-HRM practices and organizational outcomes suggests the need for policy levers that promote the adop-tion of environmentally responsible HR practices by organizations. The encouragement of sustainability initiatives in HR policies can contribute national achievement of, for exam-ple, Sustainable Development Goals (SDGs) and Conférence des Parties (COP) goals. In addition, policymakers can support research programs aimed improving and under-standing the implementation of G-HRM practices across different sectors and cultural contexts, facilitating evidence-based policy development.

Reviewer 2 Report

Comments and Suggestions for Authors

This research, supported by a literature review, highlights that Green Human Resource Management (G-HRM) practices foster environmentally conscious attitudes and behaviors among employees, contributing to enhanced employee satisfaction and improved organizational ecological performance.

The study follows a logical structure and provides valuable information. Nevertheless, from the reviewer's perspective, the author(s) should address some issues to improve the submission quality. Please see the detailed comments.

C1. In the introduction section, the author(s) should emphasize the relevance of Sustainability and the urgency to adopt sustainable models and practices. See the flowing non-mandatory suggestions: "As emphasized by the United Nations General 17 Sustainable Development Goals (SDGs), organizations need to operationalize and integrate Sustainability in their strategy, plans, and actions. Therefore, they can address the current and forthcoming stakeholder needs and ensure a better and sustainable future for all, balancing economic, social, and environmental development (Fonseca et al., 2020). Moreover, academic research (Leal Filho et al., 2023) highlights the need for stronger action to avoid major challenges such as a climate catastrophe".

References:

·         Fonseca, L.M., Domingues, J.P. and Dima, A.M. (2020). Mapping the Sustainable Development Goals Relationships. Sustainability 2020, 12, 3359; doi:10.3390/su12083359.

·         Leal Filho W, Viera Trevisan Laí, Simon Rampasso I, Anholon R, Pimenta Dinis MA, Londero Brandli L, Sierra J, Lange Salvia A, Pretorius R, Nicolau M, Paulino Pires Eustachio João Henrique, Mazutti J, When the alarm bells ring: Why the UN sustainable development goals may not be achieved by 2030, Journal of Cleaner Production (2023), doi: https://doi.org/10.1016/j.jclepro.2023.137108

C2. The author(s) should reinforce the methodological dimension of the literature review (e.g., "A literature review aims to reflect the state of knowledge in a specific subject supported by a methodical behavior (Tranfield et al., 2003)) and the choices of the selected keywords, considering the five proposed research questions. Moreover, please carefully explain the choice of the selected keywords. The keywords do not include terms such as "employee green behavior" or "client satisfaction", which might be a reason for the "scarcity of studies specifically examining the influence of G-HRM practices on employee and client satisfaction ". For the reviewer, this issue must be explained by the author(s).

Reference:

·         Tranfield, D., Denyer, D. & Smart, P. (2003). Towards a Methodology for Developing Evidence-Informed Management Knowledge by Means of Systematic Review. British Journal of Management, 14: 207-222. https://doi.org/10.1111/1467-8551.00375.

C3. The author(s) claim in the introduction that this study has a special focus on the public sector, but this is not evident in the title and the remainder of the article.

C4. Please clarify the method and criteria for studying the selected papers and how reliability and consistency were ensured.

C5. As a suggestion for future investigations, the author (s) could consider the influence of the country's culture and SDGs performance on the research results.

C6. As a research limitation, the author (s) could add that the literature review did not encompass studies from Africa, America, and Oceania.

C7. Please develop the academic, managerial and policy implications for this study.

The reviewer hopes that this feedback will be valuable to the author(s) and wishes them the best of success.

Author Response

Reviewer 2:

Comment:

There is only one recommendation: in my opinion, it is recommended to better present (some additional information and examples, if possible) for the criteria used to assess the relevance of the selected literature.

Response:

Thank you for bringing in this crucial point. We addressed your concern by expanding the methods section to include details about the study selection criteria and the systematic approach we employed to ensure quality, transparency, and rigour of our review. Please refer to the section 2.2 Study selection in line 129.

Round 2

Reviewer 2 Report

Comments and Suggestions for Authors

The author(s) replied to the reviewer's feedback, improving the manuscript accordingly, which is appreciated. Hence, this submission has no relevant objections from the reviewer's perspective. It may be suitable for publication in its present form.